# LED Light Irradiations Differentially Affect the Physiological Characteristics, Ginsenoside Content, and Expressions of Ginsenoside Biosynthetic Pathway Genes in *Panax ginseng*

Ping Di, Zhuo Sun, Lin Cheng, Mei Han, Li Yang and Limin Yang *

Key Laboratory by Province and the Ministry of Science and Technology of Ecological Restoration and Ecosystem Management, College of Chinese Medicinal Material, Jilin Agricultural University, Changchun 130118, China; yangli@jlau.edu.cn (L.Y.)
* Correspondence: yanglimin@jlau.edu.cn; Tel.: +86-133-8430-9292

**Abstract:** Light is essential for plants and plays a vital role in their growth and development. Light irradiation affects the physiological characteristics and synthesis of secondary metabolites in plants. As a semi-shade perennial plant, *Panax ginseng* C.A. Mey. is sensitive to changes in the light environment. Different light irradiations significantly affect the secondary metabolic processes of *P. ginseng*. However, few studies have investigated the changes in ginsenoside content in *P. ginseng* under different light irradiation conditions. In this study, 3-year-old *P. ginseng* was cultured under white (CK) light, blue (B) light, red (R) light, green (G) light, and natural light (NL) to explore the effects of light irradiation on the physiological characteristics and ginsenoside secondary metabolism of *P. ginseng*. The B and CK treatments significantly increased the photosynthetic level in *P. ginseng* leaves. The total saponin content under blue and red light treatments increased by 28.81% and 21.64%, respectively, compared with the CK treatment. Blue and red light improved the transcription levels of ginsenoside biosynthetic pathway genes. Blue light upregulated the expression of *HMGR*, *SS*, *SE*, *DS*, *CYP716A52*, and *CYP716A47*, and the expression of *HMGR*, *SS*, *SE*, *DS*, and *CYP716A47* under red light treatment was significantly upregulated in *P. ginseng* roots. Principal component and correlation analyses revealed that the physiological and ecological processes of *P. ginseng* exhibited different responses to light irradiation. The total saponin content in the roots was positively correlated with the content of protopanaxatriol -type ginsenosides and water use efficiency in leaves. Our study indicates that light conditions can be improved by blue and red light or by blue and red film covering to facilitate the accumulation of saponin during the ecological cultivation of *P. ginseng*.

**Keywords:** light irradiation; photosynthetic changes; ginsenosides content; ginsenosides biosynthesis relative gene expression



## 1. Introduction

*Panax ginseng* C.A. Meyer (*P. ginseng*) is a perennial herb (Araliaceae family) predominantly distributed in the northeastern region of China, Korea, and the eastern region of Siberia. As a traditional Chinese medicinal plant, *P. ginseng* has many medicinal effects, including anticancer, antidiabetic, and antioxidant effects [1–3]; additionally, it stimulates immune function [4], enhances cardiovascular health [5], suppresses anxiety [6], and improves memory and learning [7]. Over the years, the cultivation model of *P. ginseng* has mainly used deforestation to plant *P. ginseng* in China. Forestry resources are overexploited, and forest ecosystems continue to lose balance in this model [8]. Therefore, the Chinese government introduced a policy to preserve forestland. The model of *P. ginseng* cultivation gradually changed from forestland planting to farmland planting. Establishing the quality of *P. ginseng* under the farmland cultivation model is the key to achieving a healthy and sustainable development of *P. ginseng* production. Therefore, scientific and technological workers have researched soil moisture, temperature, and light factor control, which

could improve the quality and yield of *P. ginseng* through rational ecological factor control modes [9–11]. Presently, research on this remains insufficient; therefore, formulating effective and standardized cultivation measures and exploring new cultivation modes for *P. ginseng* are urgent problems to be solved for the healthy and long-term development of the *P. ginseng* industry.

The pharmacodynamic components of medicinal plants are closely related to light, temperature, soil, and other ecological factors [12]. Light is a crucial ecological component that profoundly influences plant development, growth, and secondary metabolism. Light intensity, quality, and photoperiod affect plant energy metabolism and dry matter accumulation. Additionally, light is the signal capable of regulating plant physiological and ecological processes, and plants have complex light signal transduction modes. For example, plants can transport light signals from above-ground parts to roots by direct transmission, hormone transport, and protein transfer [13]. Light quality affects the morphological and physiological changes in plants [14–16]. Red light drives photosynthesis through photosynthetic pigments; it promotes C metabolism and carbohydrate synthesis in the leaves of plants [17]. Blue light inhibits stem elongation [18], regulates stomatal opening and plant phototropism [19,20], and affects plant photomorphogenesis through blue light and UV photoreceptors [21]. In addition, blue light improves plant protein content and promotes N metabolism [22]. Green light is rarely absorbed in chloroplasts, and most of the light is reflected by the leaves; therefore, green light is considered an ineffective light source [23]. Notably, green light also promotes photosynthesis in plants. Green light reaches the lower canopy of plants and increases photosynthesis in the lower canopy where light is lacking [24]. Cao et al. confirmed that ginger presented shade avoidance symptom characteristics and upregulation of chlorophyll metabolism genes after supplementing with green light under direct light. Green light increases crown volume [25].

Furthermore, several studies have reported that light irradiations positively influence the accumulation of pharmacodynamic components in medicinal plants. Lopes reported that light quality changed the histochemical, anatomical, and volatile traits of *Artemisia annua* L. Blue light has been shown to induce *ADS* gene expression and increase artemisinin content [26]. Light quality affects flavonoid content in *Cyclocarya paliurus*; blue light promotes the synthesis of flavonoids by up-regulating the expression of key enzymes in the biosynthesis pathway of flavonoids [27]. Therefore, it is necessary to regulate the light environment of *P. ginseng* as this can improve the yield of *P. ginseng* and ensure that the medicinal qualities of *P. ginseng* are stable and controllable.

Notably, several studies have demonstrated that ginsenosides are the main bioactive components and pharmacodynamic material in *P. ginseng* [28]. Ginsenosides are a series of glycosylated triterpenoids, which can be divided into dammarane- and oleanane (OA)-type saponins, and dammarane-type ginsenosides, including protopanaxadiol (PPD)-, protopanaxatriol (PPT)-, and ocotillol (OCT)-type, according to their skeletons [29]. The mevalonate (MVA) pathway plays a dominant role in the synthesis of triterpene saponins [30]. The synthetic pathway can be divided into three stages: (1) acetyl-CoA forms isopentenyl pyrophosphate (IPP), and its isomer, dimethylallyl pyrophosphate (DMAPP) through the mevalonate (MVA) pathway; (2) IPP and DMAPP are converted to 2,3-oxysqualene; and (3) 2,3-oxysqualene cyclization, hydroxylation, and glycosylation of functional groups on the ring [31]. *HMGR* is a key enzyme located upstream of the ginsenoside synthesis pathway that regulates ginsenoside biosynthesis by influencing the generation of IPP and DMAPP [32]. Farnesyl pyrophosphate synthase (*FPS*), squalene synthase (*SS*), and squalene epoxidase (*SE*) are midstream genes that determine triterpenoid saponin synthesis [33,34]. Xu et al. reported that *FPS* may contribute the most to the synthesis or accumulation of total saponins in the stems, followed by the *SS* and *DS* (dammarenediol-II synthase) [35]. Cytochrome P450s (*CYPs*) are one of the largest families of enzymatic proteins that promote the structural diversity of ginsenosides, including *CYP716A47*, *CYP716A53v2*, *CYP716A52v2*, etc. [36,37]. Although the pathway of ginsenoside synthesis has been studied extensively, how light quality affects the transcription of ginsenoside synthesis genes and changes

in ginsenosides content remains unknown; this limits further development of *P. ginseng* cultivation using manually controlled light irradiation.

At present, there are few studies on the mechanism of ginsenoside synthesis under different light irradiations. Our study aimed to investigate how light irradiation influences physiological changes, ginsenoside accumulation, and the expression of key genes in the ginsenoside biosynthesis pathway in *P. ginseng*; to clarify the mechanism by which light irradiation changes the synthesis of ginsenosides in *P. ginseng*; and determine the light irradiations that promote ginsenoside accumulation. The findings of this study are predicted to be extremely useful in better understanding the physiological responses to diverse light situations in *P. ginseng* and provide a theoretical basis for innovation in the ecological cultivation technology of *P. ginseng*.

## 2. Materials and Methods

### 2.1. Experimental Materials

*P. ginseng* seedlings were collected from the medicinal plant garden of Jilin Agricultural University and identified as *P. ginseng* seedlings (3 years old) by Professor Li-min Yang of Jilin Agricultural University. Three-year-old seedlings that were robust, disease-free, and strong were transferred. Polypropylene pots (24 cm (diameter) × 16 cm (height)) were used to transplant *P. ginseng.* Two ginseng plants were placed per pot filled with 4.5 kg of soil. The basic requirements of *P. ginseng* cultivated soil included pH 5.7, 1.32 mg/Kg organic carbon, 287.41 mg/kg available nitrogen, 14.13 mg/kg available phosphorus, and 152.44 mg/kg available potassium. Before the experiment, the pots were placed in a greenhouse under 10% shading to ensure the normal growth of *P. ginseng*. Water, manure, and pest control management were administered before the experiment. The light irradiation regulation commenced when *P. ginseng* grew to the red fruit stage.

### 2.2. Light Treatment

The experimental site was located at the agricultural base facility A03 of the Jilin Agricultural University, China. On 18 August 2020, *P. ginseng* seedlings with the same growth status were selected. Eight pots of *P. ginseng* were moved to the culture room and placed on a cultivation rack. The cultivation rack was covered with a silvery-silver shading cloth to isolate the external light source. A ventilation fan with a power of 34 W was installed to maintain air exchange and temperature balance inside and outside the culture room. In this study, light-emitting photodiodes (LEDs; Shenzhen Chenhua Energy Efficient Lighting Co., Shenzhen, China) were used as the light sources. White LED light (450 nm–660 nm) was used as the control treatment (CK), while the other treatments included blue light (444 nm), red light (660 nm), green light (515 nm), and natural light (NL) treatment (*P. ginseng* growth in a greenhouse with 10% shading). The spectra and intensities of the LEDs were measured using a spectral color illuminance measurement system (PLA-20, Hangzhou Everfine Optoelectronic Information Co., Ltd., Hangzhou, China) and are shown in Figure 1 The photosynthetic photon flux density of the *P. ginseng* canopy was maintained at $80 \pm 10$ µmol m$^{-2}$ s$^{-1}$. The *P. ginseng* pots were rotated every three days to maintain a uniform distribution of the irradiated light, and the light period was set at 12 h/12 h. The temperature of the culture room was set at 25 °C/20 °C. Routine management practices were implemented. After 30 d of light treatment, *P. ginseng* root, stem, and leaf samples were collected to measure various indices. The samples were divided into three biological repetitions for the next index determination. Some of the fresh samples were promptly frozen in liquid nitrogen and kept at −80 °C. The remaining samples were dried at 50 °C to determine their ginsenoside concentration.

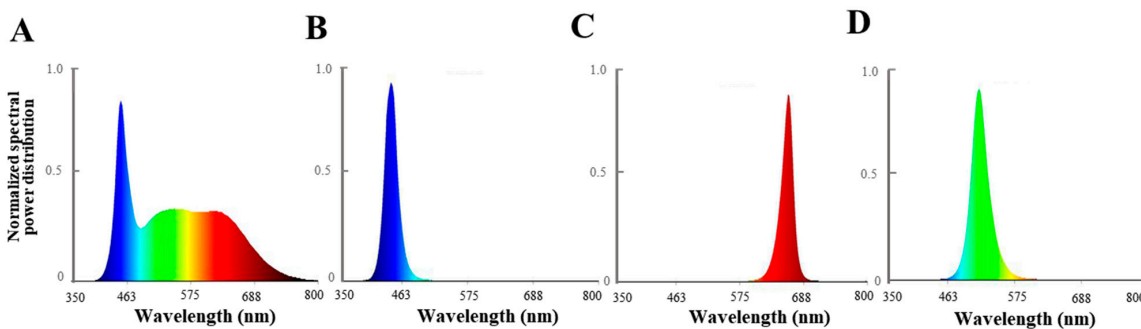

**Figure 1.** Spectral distribution of different light irradiation treatments: (**A**) white LEDs; (**B**) blue LEDs; (**C**) red LEDs; (**D**) green LEDs.

### 2.3. Determination of Chlorophyll Content of P. ginseng Leaves

The chlorophyll content was estimated using Porra's method [38]. Total chlorophyll was extracted using 80% acetone. The extract was analyzed using a spectrophotometer (Spectra max 190, Molecular Devices Co., Ltd., San Francisco, CA, USA) with 470, 645, and 663 nm light absorption values.

### 2.4. Determination of Photosynthetic Parameters

Five plants and three leaves from each plant were selected to measure the photosynthesis rate (Pn), transpiration rate (Tr), stomatal conductance (Gs), intercellular $CO_2$ concentration (Ci), and atmospheric $CO_2$ concentration (Ca) by a portable photosynthesis instrument (LCpro+, ADC BioScientifc Co., Ltd., Hoddesdon, UK) from 9:00 a.m. to 11:00 a.m. The light source of the portable photosynthesis instrument was used to ensure consistency of light intensity, which was set at 80 μmol m$^{-2}$ s$^{-1}$ under the various treatments. The formulas Ls = 1 − Ci/Ca and WUE = *Pn/Tr* were used to calculate the stomatal limit value (Ls) and instantaneous water-use efficiency (WUE). A chlorophyll fluorescence instrument (OS-5P+, Opti-Sciences Co., Ltd., Boston, MA, USA) was used to measure the chlorophyll fluorescence parameters of *P. ginseng* leaves. Maximum photochemical efficiency (*Fv/Fm*), actual photochemical efficiency (φPSII), photochemical quenching coefficient (qP), and non-photochemical quenching coefficient (NPQ) were measured and recorded [39].

### 2.5. Determination of Content of the Photosynthetic Products of P. ginseng

A total of 0.5 g of fresh tissue samples and 4.5 mL PBS (0.05 mol/L) were mixed and ground. After centrifugation for 10 min (4500 rpm/min) in an ice water bath, the supernatant was collected for testing. The homogenate of tissues was treated with soluble sugars, sucrose, glucose, and starch kits (Nanjing Jiancheng Bioengineering Institute, Nanjing, China). The absorbance of the reaction solution was measured at 620, 505, and 620 nm using an enzyme-labeling instrument to determine the soluble sugar, glucose, and starch contents (Spectra max 190, Molecular Devices Co., Ltd., San Francisco, CA, USA); the absorbance of the reaction solution was measured at 290 nm using a UV spectrophotometer (Evolution 201, Thermo Scientific Co., Ltd., Waltham, MA, USA) to determine the sucrose content. The protein content was calculated according to the manufacturer's instructions.

### 2.6. Determination of Total Saponin and Ginsenoside Content

Ginsenosides were extracted using the microwave-assisted method (Mars6, CEM Co., Ltd., Boston, MA, USA). An amount of 1.0 g *P. ginseng* root and leaf powder was added to 20 mL 80% methanol, and extraction was performed for 5 min at 45 °C, 600 W. This process was performed three times, and the extract solution was filtered. The solution was concentrated, and the volume was reduced to 5 mL.

The total saponin content was measured using $H_2SO_4$ and vanillin, with minor modifications. Exactly 50 μL of the extract was evaporated using a water bath in a test tube.

More precisely, 0.5 mL of 8% vanillin and 5 mL of 72% $H_2SO_4$ were added to the test tube and mixed thoroughly. The test tubes were immersed in a water bath at 60 °C for 10 min. The test tubes were then transferred to an ice-water bath for 10 min. The samples were analyzed using a spectrophotometer at 554 nm light absorption. A total amount of 0.5 mL of 8% vanillin and 5 mL of 72% $H_2SO_4$ were accurately added to the test tube and carefully mixed. The test tubes were then placed in a cold bath for 10 min. A spectrophotometer was used to measure light absorption at 554 nm in the samples.

Ginsenoside content was measured by an HPLC system (1260 Infinity II, Agilent Technologies Co., Ltd., Santa Clara, CA, USA). A reverse phase C18 column (ZORBAX SB-C18, 4.6 × 250 mm, 5 μm, Agilent, Santa Clara, CA, USA) was used. The 0.22 μm organic phase filter was used to filter extracting. The mobile phase consisted of A (100% water) and C (100% acetonitrile). The injection volume was 10 μL, and the system was operated at 25 °C. The flow rate was 0.8 mL/min, and the detection wavelength was 203 nm. The gradient program was as follows: 0–36 min, 18–21% C; 37–41 min, 21–28% C; 41–45 min, 28–34% C; 45–54 min, 34–38% C; 54–61 min, 38–71% C; 61–80 min, 71–80% C; and 80–100 min, 80–18% C. Commercial-grade ginsenosides ($Rg_1$, Re, Rf, $Rb_1$, $Rb_2$, Rc, Rd, and $Rb_3$) were purchased from Shanghai Source Leaf Biological Technology Co. Ltd. A calibration curve was constructed to calculate ginsenoside content under different light conditions. The standard curves were established using the ginsenosides standard sample ($Rg_1$ y = 4623.1x − 3.6071, Re y = 3435x − 8.2564, Rf y = 5185.4x + 17.907, $Rb_1$ y = 3539.8x + 12.262, Rc y = 3841.7x + 9.3639, $Rb_2$ y = 3356.5x − 4.5932, $Rb_3$ y = 2791.6x + 25.263, and Rd y = 3763x + 20.817).

### 2.7. Key Genes Involved in the Ginsenoside Synthesis Pathway

Total RNA was extracted using an RNA plant extraction kit (Beijing Zoman Biotechnology Co., Ltd., Beijing, China). RNA purity and concentration were determined using a P330 nanophotometer (Thermo Fisher Scientific, Waltham, MA, USA). RNA samples were obtained with absorption ratios of $A_{260/280} > 1.8$ and $A_{260/280} < 2.2$. Total RNA was transcribed into cDNAs using a reverse transcription kit (TaKaRa Biotechnology Co., Ltd., Beijing, China).

The relative gene expression levels of HMGR, FPS, SS, SE, DS, CYP716A47, CYP716A52v2, and CYP716A53v2 were determined using GAPDH as an internal reference (Table 1 presents the primer information for the studied genes).

**Table 1.** PCR primers used for the quantification of gene expression levels by qRT-PCR.

| Gene | Primer Sequence 5–'3' |
|---|---|
| GAPDH | F: ATGGACCATCAGCAAAGGAC R: GGTAGCACTTTCCCAACAGC |
| HMGR | F: TTGCGGGTCCATTGCTGCT R: CTCTGGTCATCCCATCTTTTA |
| FPS | F: CAAGAAGCATTTCCGACAA R: CTCTCCTACAAGGGTGGTGA |
| SS | F: GGACTTGTTGGATTAGGGTTG R: ACTGCCTTGGCTGAGTTTTC |
| SE | F: ATGCTTTGAATATGCGCCATC R: CATGGAGATCGCGTAAAGGTC |
| DS | F: ATAGGGCAATGATAAGGGGAG R: ACCGCCGTTGAGATTAGATG |
| CYP716A47 | F: TCACCTTCGTTCTCAACTATC R: TCTTCCTCAAATCCTCCCAAT |
| CYP716A52v2 | F: AGGAGCAAATGGAGATAG R: AACCGTTGTAGGTGAAAT |
| CYP716A53v2 | F: ATCGGACAACGAGGCAGCAC R: GCCAACAGGCCAACTCAA |

GAPDH, glyceraldehyde-3-phosphate dehydrogenase; HMGR, 3-hydroxy-3-methylglutaryl-coenzyme A reductase; FPS, farnesyl pyrophosphate synthase; SS, squalene synthase; SE, squalene epoxidase; DS, dammarenediol-II synthase; CYPs, cytochrome P450.

The reaction conditions were as follows: 30 s at 94 °C for pre-denaturation, 5 s at 95 °C for denaturation, 32 s at 55 °C for annealing, and 20 s at 72 °C for extension. The thermal cycle was repeated 40 times. An Mx 3000 P quantitative PCR instrument (Agilent, Santa Clara, CA, USA) was used to perform the reaction.

### 2.8. Statistical Analyses

The original data were processed using Excel 2019 (Microsoft Corp., Redmond, WA, USA). All analyses were conducted using SPSS version 19 (SPSS Inc., Chicago, IL, USA). GraphPad Prism 6 (GraphPad Software Inc., San Diego, CA, USA) was used to draw the graphics.

## 3. Results

### 3.1. Fresh Weight and Dry Weight of P. ginseng Roots

The effects of different light irradiations on the fresh and dry weights of *P. ginseng* roots are presented in Figure 2. There were no significant differences between treatments for fresh and dry weights of *P. ginseng* roots after different light treatments ($p < 0.05$). Our results revealed that the underground yield of 3-year-old *P. ginseng* did not significantly change under short-term light irradiation.

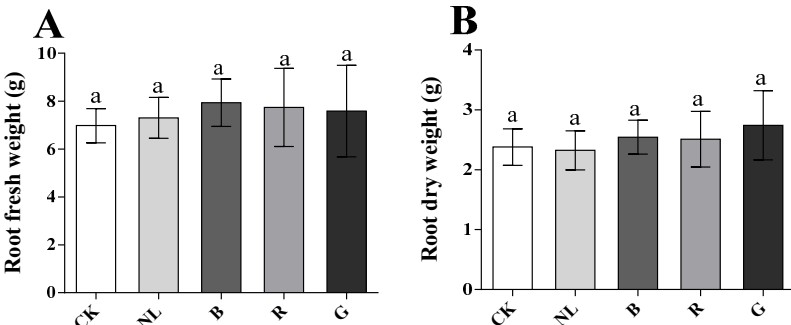

**Figure 2.** Effect of different light irradiations on root fresh weight (**A**) and root dry weight (**B**) CK, white LEDs; NL, natural light; B, blue LEDs; R, red LEDs; G, green LEDs; The data are expressed as the mean ± SD (Duncan, $p < 0.05$).

### 3.2. The Photosynthetic Pigment Content of P. ginseng Leaves

Light irradiation significantly affected the photosynthetic pigment content in *P. ginseng* leaves (Table 2). The Chl a, Chl b, and total Chl contents in leaves were the highest under blue and green light. The total Chl contents under blue and green light treatments were 1.75 mg/g and 1.74 mg/g, respectively; these values increased by 14.38% and 13.73% compared with CK ($p < 0.05$). Furthermore, compared with the red light treatment, the values increased by 37.80% and 37.01%, respectively. The carotenoid contents in B and CK treatments were the highest, 3.52 mg/g and 3.44 mg/g, respectively. Chl a/b was the highest under the blue light and CK treatments; therefore, our study revealed that blue and green lights exhibited positive effects on Chl a and Chl b contents. Additionally, blue and white light had a positive effect on carotenoid content and Chl a/b.

**Table 2.** Effect of different light irradiations on the content of chlorophyll.

| Treatments | Chl a (mg/g) | Chl b (mg/g) | Carotenoid (mg/g) | Chl a/b | Total Chl (mg/g) |
|---|---|---|---|---|---|
| CK | 1.2 ± 0.05 b | 0.33 ± 0.02 b | 0.4 ± 0.01 bc | 3.44 ± 0.17 ab | 1.53 ± 0.03 b |
| NL | 1.02 ± 0.06 c | 0.33 ± 0.01 b | 0.37 ± 0.02 cd | 3.14 ± 0.05 cd | 1.35 ± 0.07 c |
| B | 1.37 ± 0.06 a | 0.39 ± 0.02 a | 0.47 ± 0.02 a | 3.52 ± 0.06 a | 1.75 ± 0.08 a |
| R | 0.95 ± 0.07 c | 0.32 ± 0.03 b | 0.34 ± 0.02 d | 3.02 ± 0.1 d | 1.27 ± 0.09 c |
| G | 1.31 ± 0.1 ab | 0.43 ± 0.05 a | 0.43 ± 0.02 b | 3.28 ± 0.13 bc | 1.74 ± 0.12 a |

Note: Chl a: chlorophyll a; Chl b: chlorophyll b; Total Chl: total chlorophyll; CK, white LEDs; NL, natural light; B, blue LEDs; R, red LEDs; G, green LEDs. The data are expressed as the mean ± SD (Duncan, $p < 0.05$). The different lowercase letters indicate significant differences between treatments.

### 3.3. The Photosynthetic Parameters of P. ginseng Leaves

To explore the effect of different light qualities on *P. ginseng* leaves, we investigated the effect of light irradiations on the photosynthesis of *P. ginseng* leaves. Changes in the photosynthetic parameters of *P. ginseng* leaves under different light treatments are presented in Figure 3. The $P_n$ significantly increased to 1.91 and 1.73 $\mu mol/m^2/s$ under the CK and blue light treatments compared with the other treatments ($p < 0.05$). The $P_n$ of leaves was lower than that of the other treatments under green and natural light treatments, decreasing by 53.20% and 62.17%, respectively, compared with the CK treatment. There were no significant differences in WUE under different light treatments. The $G_s$ of leaves exhibited the same trend as the $P_n$, indicating that the $G_s$ of the CK and blue light treatments had the highest red light, followed by the $G_s$ of green and natural light. The $L_s$ of the blue light and CK treatments were significantly lower than those of the other treatments ($p < 0.05$), being 24.20% and 29.94% lower than that of the green light, respectively. The results revealed that the photosynthetic rate of *P. ginseng* leaves was higher under the blue light and CK treatments, followed by red light. Stomatal conductance was lower under green and natural light treatments, and the stomatal limit value was higher. Photosynthesis in *P. ginseng* leaves was limited under green and natural light treatments.

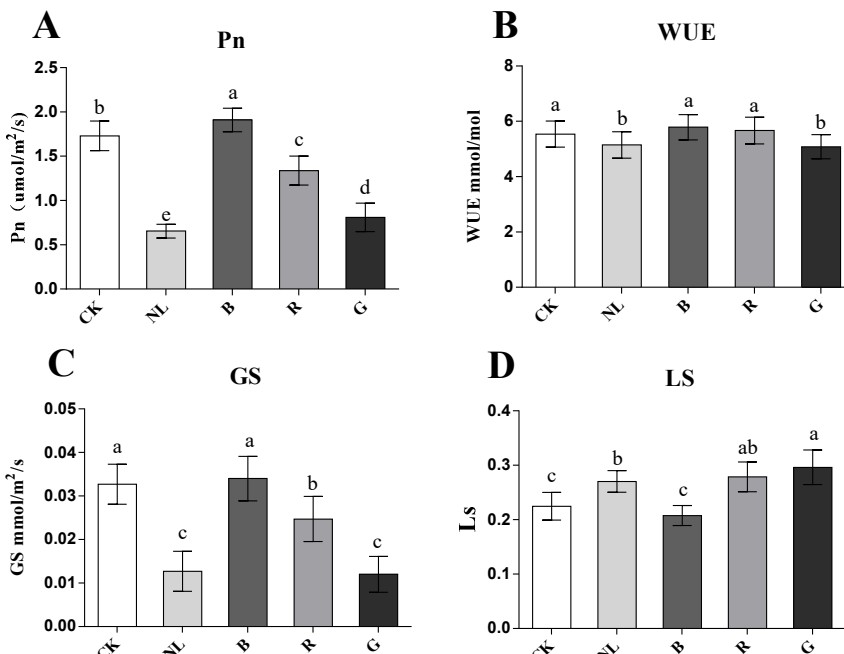

**Figure 3.** Effect of different light irradiations on photosynthesis rate (Pn) (**A**), water-use efficiency (WUE) (**B**), stomatal conductance (Gs) (**C**), and stomatal limit value (Ls) (**D**) in *P. ginseng* leaf. CK, white LEDs; NL, natural light; B, blue LEDs; R, red LEDs; G, green LEDs. The data are expressed as the mean ± SD (Duncan, $p < 0.05$). The different lowercase letters indicate significant differences between treatments.

### 3.4. The Chlorophyll Fluorescence Parameters of P. ginseng Leaves

Light irradiation exhibited a significant effect on the chlorophyll fluorescence parameters in *P. ginseng* leaves (Figure 4). The *Fv/Fm* ratio reflects the ability of plants to accept light energy to mediate photochemical reactions and reflects the activity of photosystem II. The *Fv/Fm* was higher under the CK, green, and blue light treatments and lower under the red and natural light treatments than the other treatments. The φPS II changed significantly under different light conditions. The results indicated that φPS II was higher under the CK, blue light, and green light treatments and increased by 15.44%, 29.47%, and 13.19%, respectively, compared with the natural light treatment. The qp of blue light treatment was significantly higher than the other light treatments ($p < 0.05$), and there was no significant

difference between the other treatments. The NPQ increased significantly under red light treatment ($p < 0.05$); however, the NPQ was the lowest under the blue light treatment. Therefore, photosystem II activity was higher under blue, white, and green light treatments in *P. ginseng* leaves. However, the reaction process of photosystem II is limited under natural light and red light treatment.

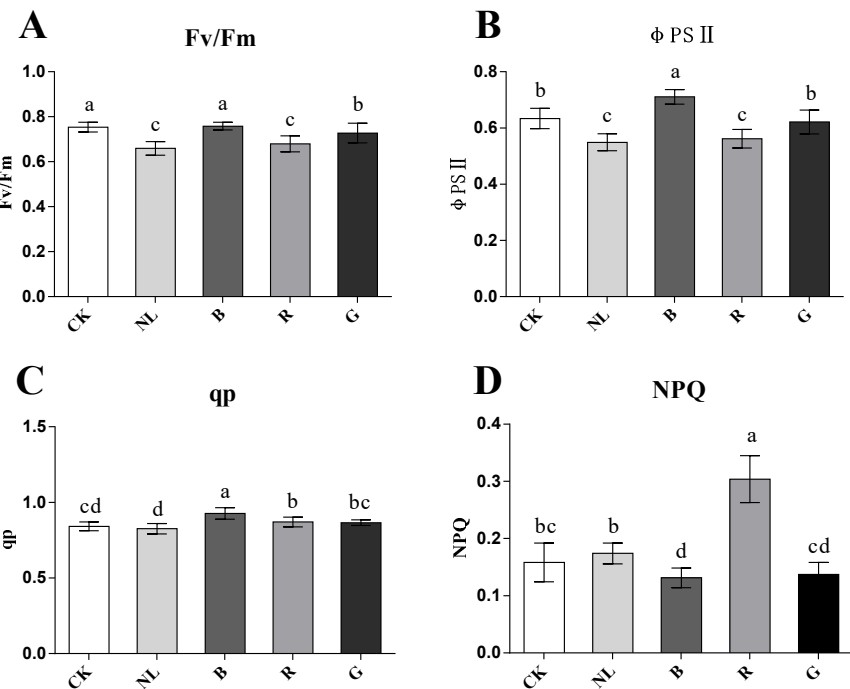

**Figure 4.** Effect of different light irradiations on maximum photochemical efficiency (*Fv/Fm*) (**A**), actual photochemical efficiency (φPSII) (**B**), photochemical quenching coefficient (qp) (**C**), non-photochemical quenching coefficient (NPQ) (**D**) in *P. ginseng* leaf. CK, white LEDs; NL, natural light; B, blue LEDs; R, red LEDs; G, green LEDs. The data are expressed as the mean ± SD (Duncan, $p < 0.05$). The different lowercase letters indicate significant differences between treatments.

### 3.5. The Photosynthate Content of P. ginseng Root and Leaves

The effect of light irradiation on photosynthate content in *P. ginseng* roots and leaves is presented in (Table 3). The content of starch in *P. ginseng* was highest under red light treatment at 495.72 mg/g and lowest under natural light treatment at 371.54 mg/g; there was little difference between the other treatments. After treatment, the contents of glucose and sucrose in *P. ginseng* roots under red light and white light treatments significantly increased compared with the other treatments; the content of glucose increased by 53.85% and 61.54% compared with natural light, the content of sucrose increased by 19.72% and 10.65% compared with natural light treatment, and the content of sucrose increased by 47.73% and 36.54%, respectively, compared with the blue light treatment. There was no significant difference in the soluble sugar content in the roots under different light irradiation treatments. In the *P. ginseng* leaves, the soluble sugar content was significantly higher under red light and natural light than in other treatments, reaching 75.17 mg/g and 72.19 mg/g and increased by 28.28% and 23.19% compared with the control treatment, respectively. The contents of starch and glucose were highest under red light treatment, increasing by 23.61% and 10.92%, respectively, compared with the control treatment. There was no significant difference in the sucrose content of the leaves under the different light irradiation treatments. Therefore, red light treatment exhibited a more positive effect on the accumulation of photosynthates in *P. ginseng* leaves.

**Table 3.** Effect of different light irradiations on photosynthate content of *P. ginseng* root and leaves.

| | Treatment | Soluble Sugar (mg/g) | Starch (mg/g) | Glucose (mg/g) | Sucrose (mg/g) |
|---|---|---|---|---|---|
| Root | CK | 18.86 ± 1.62 a | 462.52 ± 5.11 bc | 0.21 ± 0.02 a | 17.45 ± 0.25 a |
| | NL | 19.61 ± 0.74 a | 371.54 ± 10.49 d | 0.13 ± 0.01 b | 15.77 ± 1.21 b |
| | B | 18.96 ± 1.26 a | 477.46 ± 4.15 ab | 0.11 ± 0.01 b | 12.78 ± 0.61 c |
| | R | 19.77 ± 0.7 a | 495.72 ± 5.49 a | 0.2 ± 0.02 a | 18.88 ± 1.11 a |
| | G | 20.79 ± 2.99 a | 442.27 ± 26.95 c | 0.11 ± 0.01 b | 12.33 ± 0.43 c |
| Leaf | CK | 58.6 ± 0.85 c | 49.42 ± 3.52 c | 4.58 ± 0.15 b | 25.05 ± 1.43 a |
| | NL | 72.19 ± 3.86 a | 54.22 ± 1.86 bc | 4.24 ± 0.2 bc | 24.38 ± 0.77 a |
| | B | 61.29 ± 2.13 bc | 58.2 ± 3.74 ab | 3.29 ± 0.31 d | 24.07 ± 0.78 a |
| | R | 75.17 ± 1.01 a | 61.09 ± 0.63 a | 5.08 ± 0.16 a | 23.23 ± 1.24 a |
| | G | 63.6 ± 3.5 b | 49.56 ± 3.04 c | 4.12 ± 0.3 c | 23.08 ± 0.46 a |

Note: CK: white LEDs, NL: natural light, B: blue LEDs, R: red LEDs, G: green LEDs. The data are expressed as the mean ± SD (Duncan, $p < 0.05$). The different lowercase letters indicate significant differences between treatments.

### 3.6. Ginsenoside Content of P. ginseng Roots, Leaves, and Stems

Ginsenosides are important secondary metabolites and are major pharmacodynamic components. Our study determined the ginsenoside content of *P. ginseng* roots, stems, and leaves under different light irradiation conditions. We investigated the response characteristics of light irradiation on ginsenosides in various tissues of *P. ginseng*. As presented in Figure 5, the saponin content in the ginseng root changed significantly under different light irradiations. Blue and red light treatments increased the content of ginsenosides Rg1, Re, Rb1, Rc, and Rb3, in which ginsenosides Rg1, Re, and Rb1 were the index components specified by the Chinese Pharmacopoeia. The contents of Rg1, Re, and Rb1 reached 1.67 mg/g, 4.20 mg/g, and 1.45 mg/g in *P. ginseng* root under blue light treatment, representing increases of 40.73%, 68.52%, and 18.31% compared with white light treatment, respectively. The contents of Rg1, Re, and Rb1 reached 1.45 mg/g, 3.83 mg/g, and 1.40 mg/g in *P. ginseng* root under red light treatment, representing increases of 22.59%, 53.80%, and 14.28% compared with white light treatment, respectively. Compared with other treatments, PPD type, PPT type, and total saponin content increased under blue light and red light treatments. The total saponin content under blue light treatment increased by 28.81% compared with white light, and the total saponin content increased by 21.64% under red light treatment compared with white light.

The effects of different light irradiations on the saponin content in *P. ginseng* leaves are presented in Figure 6. Blue and red light treatments decreased the content of ginsenosides Rg1 in *P. ginseng* leaves to 10.96 mg/g and 9.29 mg/g, respectively, representing decreases of 11.30% and 24.82% compared with CK. *P. ginseng* treated with natural light showed the highest ginsenoside Rb1 content (1.27 mg/g DW). The content of ginsenosides Rc, Rb2, and Rb3 showed little difference among the different light treatments. The contents of PPD, PPT, and total saponins were calculated according to the contents of ginsenosides in *P. ginseng* leaves. We discovered that the contents of PPD, PPT, and total saponins under different light treatments were not significantly different. However, there was a tendency for PPD type, PPT type, and total saponins under blue light and red light treatments to be slightly higher than in other treatments.

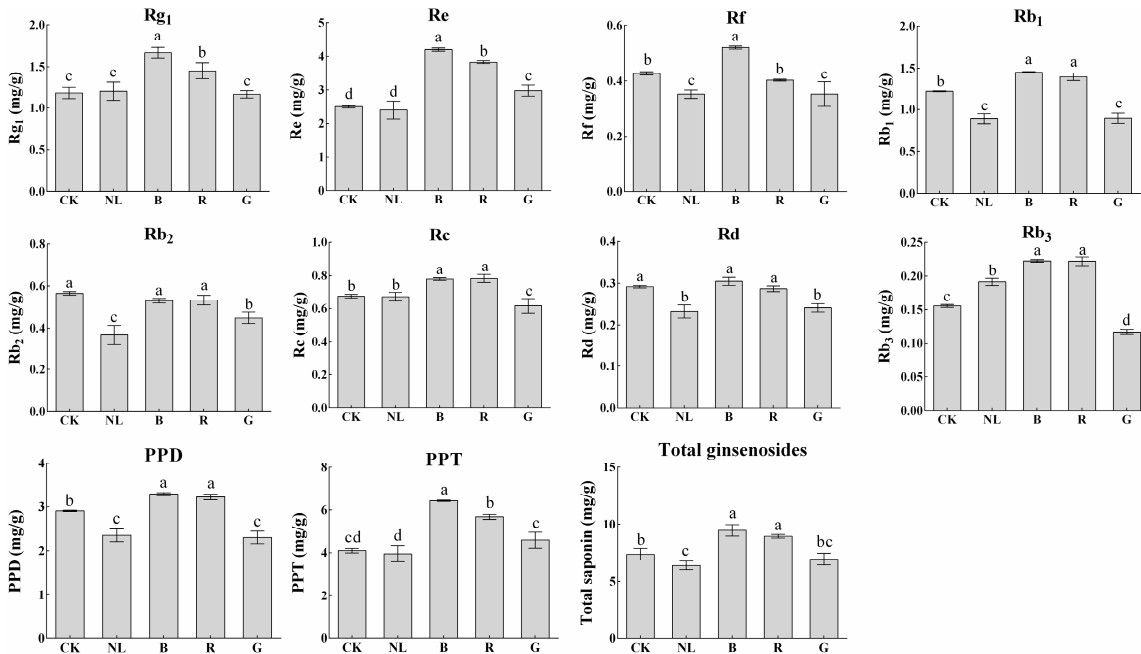

**Figure 5.** Ginsenoside contents (mg/g) of *P. ginseng* root under different light irradiations. CK, white LEDs; NL, natural light; B, blue LEDs; R, red LEDs; G, green LEDs. The data are expressed as the mean ± SD (Duncan, $p < 0.05$). The different lowercase letters indicate significant differences between treatments.

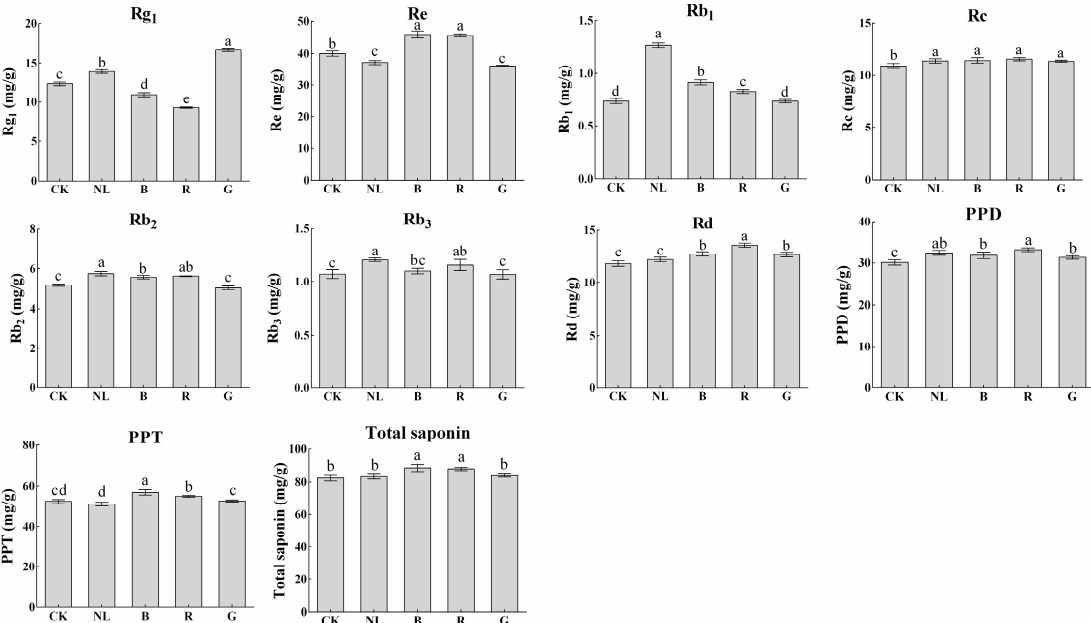

**Figure 6.** Ginsenoside contents (mg/g) of *P. ginseng* leaf under different light irradiations. CK, white LEDs; NL, natural light; B, blue LEDs; R, red LEDs; G, green LEDs. The data are expressed as the mean ± SD (Duncan, $p < 0.05$). The different lowercase letters indicate significant differences between treatments.

The effects of different light irradiations on saponin content in *P. ginseng* stems are presented in Figure 7. The content of ginsenosides was lowest under blue and red light treatments, at 2.01 mg/g and 1.75 mg/g, representing decreases of 6.76% and 18.89% compared with CK treatment, respectively. Blue and red light treatments significantly increased the content of ginsenoside Re, which was 14.85% and 7.01% higher than that of the CK treatment, respectively, and 30.19% and 21.31% higher than that of the natural light treatment, respectively. After light irradiation, the total saponin content under the blue light

treatment reached 9.00 mg/g, significantly higher than the other treatments. Our study revealed that the changes in ginsenoside Rg1, Re, Rb1, and Rd contents in *P. ginseng* leaves demonstrate a similar trend to that in stems under different light irradiation conditions. This phenomenon may be explained by abundant ginsenosides being synthesized in leaves and then transported to the stem. Notably, the synthesis and accumulation of saponins in leaves and stems are affected by light irradiation.

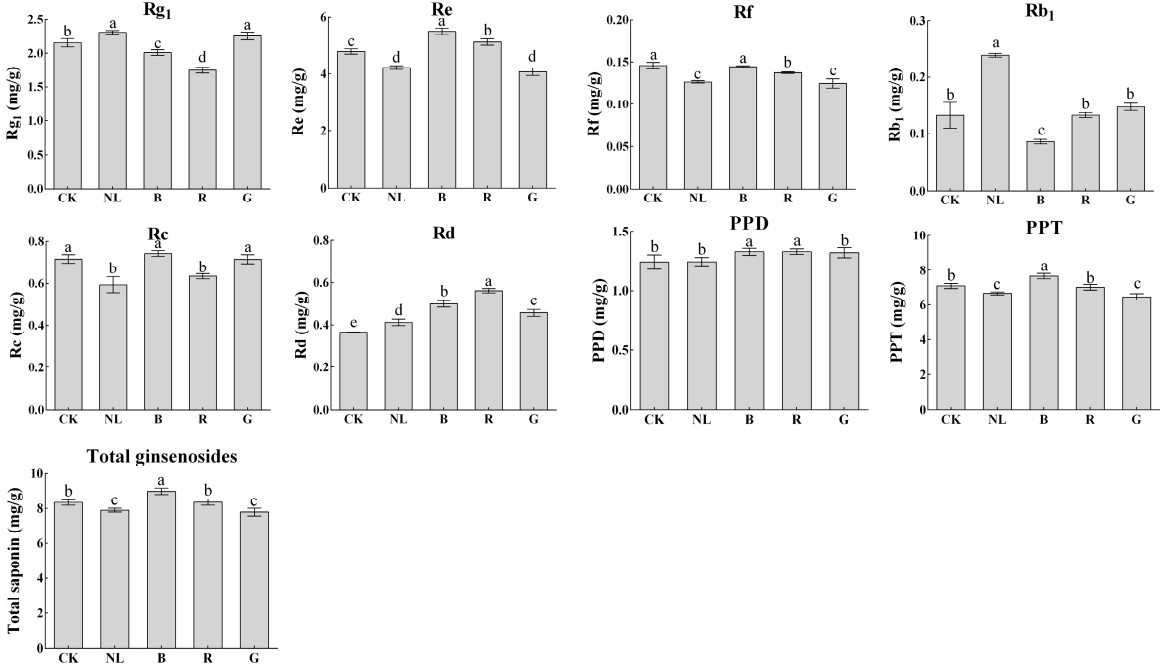

**Figure 7.** Ginsenoside contents (mg/g) of *P. ginseng* stem under different light irradiations. CK, white LEDs; NL, natural light; B, blue LEDs; R, red LEDs; G, green LEDs. The data are expressed as the mean ± SD (Duncan, $p < 0.05$). The different lowercase letters indicate significant differences between treatments.

### 3.7. PCA and Correlations between Physiological Indicators and Ginsenoside

We conducted PCA analyses to analyze the differential responses of physiological and secondary metabolites in *P. ginseng* under different light irradiation treatments and their interactive relationships. The loading and score plots of the PCA are presented in Figure 8; the results indicated that PCA1 and PCA2 explained 40.5% and 27.1% of the total variance, respectively. The distance between the point represented by the load and the origin represents the degree of contribution of that feature attribute to the principal component. We found that several indicators contributed to the principal components. Indicators such as PPD, PPT, and total ginsenosides contributed more to PCA1, and indicators such as NPQ, sucrose, and glucose contributed more to PCA2. In addition, the five light irradiation treatments were distributed in different positions in the four quadrants and were dispersed, indicating that the physiological and ecological processes of *P. ginseng* responded differently to light irradiation, and the response processes were complex and diverse.

We conducted a correlation analysis on the physiological and ecological indices and ginsenoside content of *P. ginseng* to further analyze the response mechanism of ginseng secondary metabolites to each physiological and ecological index under monochromatic light (Figure 9). The total saponin content in the roots was positively correlated with the content of PPT−type ginsenosides and leaf WUE. The PPT−type ginsenoside content in the roots was positively correlated with the PPT−type ginsenoside content and qp in the leaves. There was a significant positive correlation between the content of PPD−type ginsenosides in the roots and the total saponins in the stem. The total saponin content in the stems was positively correlated with Pn, GS, and WUE. Saponins in *P. ginseng* are

transported from shoot tissues to roots [40]. Therefore, the higher saponin content in *P. ginseng roots* under red and blue light treatments may be because blue and red light promotes the synthesis of saponins in the roots, subsequently facilitating the transport of saponins from the above-ground to the underground parts of *P. ginseng*. The synthesis and accumulation of ginsenosides responded positively to photosynthetic processes.

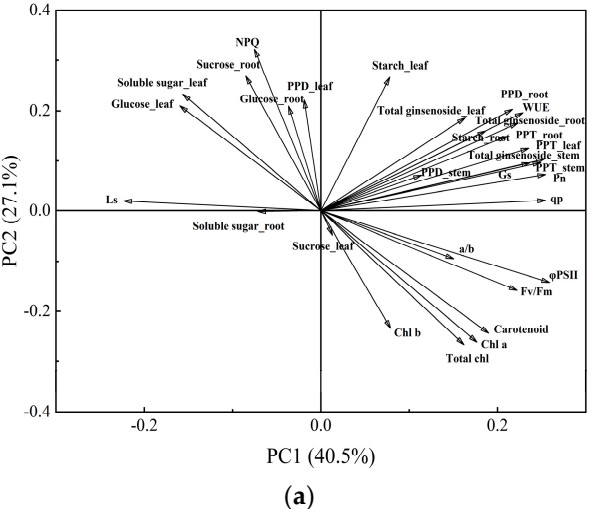

(**a**)

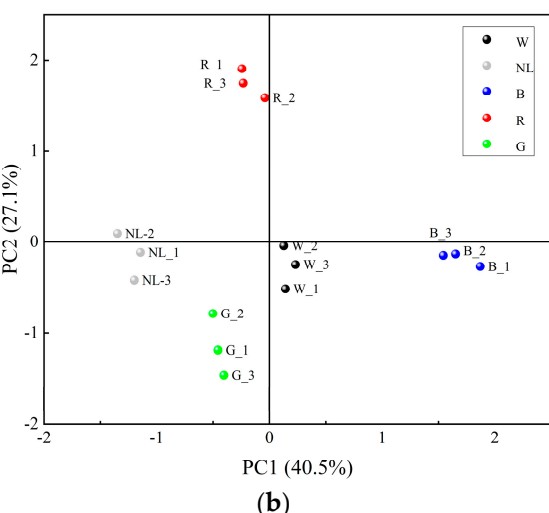

(**b**)

**Figure 8.** Principal component analysis of different light irradiations. CK, white LEDs; NL, natural light; B, blue LEDs; R, red LEDs; G, green LEDs. (**a**) Loading plots of the PCA, (**b**) Score plots of the PCA.

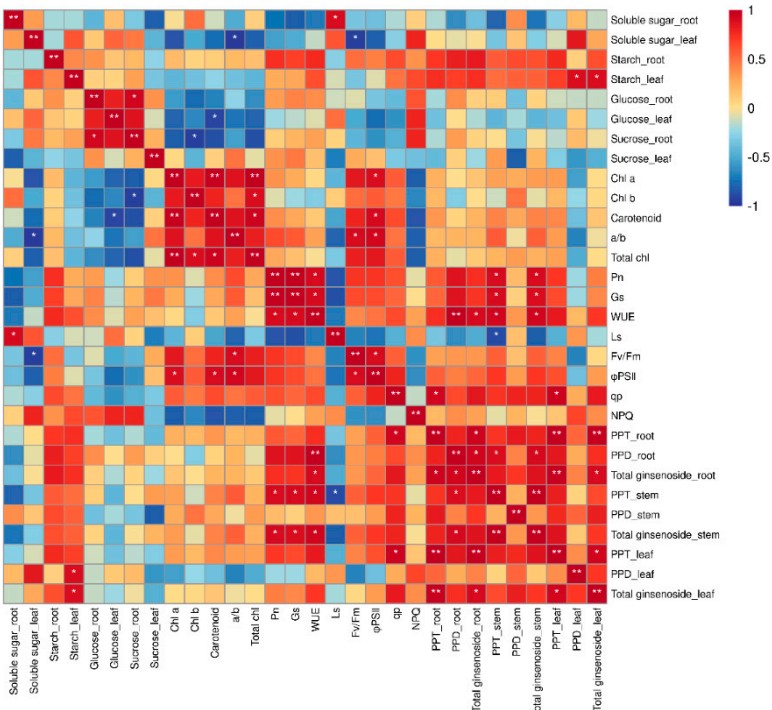

**Figure 9.** Correlation between physiological indicators and ginsenosides. * Significant at $p \leq 0.05$, ** significant at $p \leq 0.01$.

### 3.8. Key Enzyme Gene Expression in the Ginsenoside Synthesis Pathway

Since blue light and red light significantly induced ginsenoside accumulation in *P. ginseng* roots, we investigated the expression of key enzymes involved in the ginsenoside biosynthetic pathway in *P. ginseng* under CK, B, and R treatments; this was aimed at exploring the molecular mechanisms of blue and red light on saponin bioactive compound

synthesis and accumulation based on changes in gene transcription levels. The effect of light irradiation on the relative gene expression levels of *HMEGR*, *FPS*, *SS*, *SE*, *DS*, *CYP716A52*, *CYP716A53,* and *CYP716A47* in *P. ginseng* roots is presented in Figure 10. *HMGR* and *FPS* are positioned upstream of the ginsenoside synthesis pathway; *SS*, *SE*, and DS are midstream in the ginsenoside synthesis pathway. *CYP716A52*, *CYP716A53*, and *CYP716A47* function downstream of the ginsenoside synthesis pathway. The expression of *HMGR*, *SS*, *SE*, *DS*, *CYP716A52*, and *CYP716A47* in *P. ginseng* root under blue light treatment was significantly upregulated and was 1.28-, 1.46-, 1.21-, 1.57-, 1.59, and 1.43 times higher than that in the CK treatment, respectively. Additionally, the expression of *HMGR*, *SS*, *SE*, *DS*, and *CYP716A47* in *P. ginseng* root under red light treatment was significantly upregulated and was 1.92-, 1.23-, 1.15-, 1.39, and 1.29 times higher than that in the CK treatment, respectively. After blue and red light treatments, most of the key genes involved in ginsenoside synthesis were upregulated, and their expression under blue light was higher than in red light; this confirmed the phenomenon where the saponin content of *P. ginseng* root under blue light was higher than that under red light.

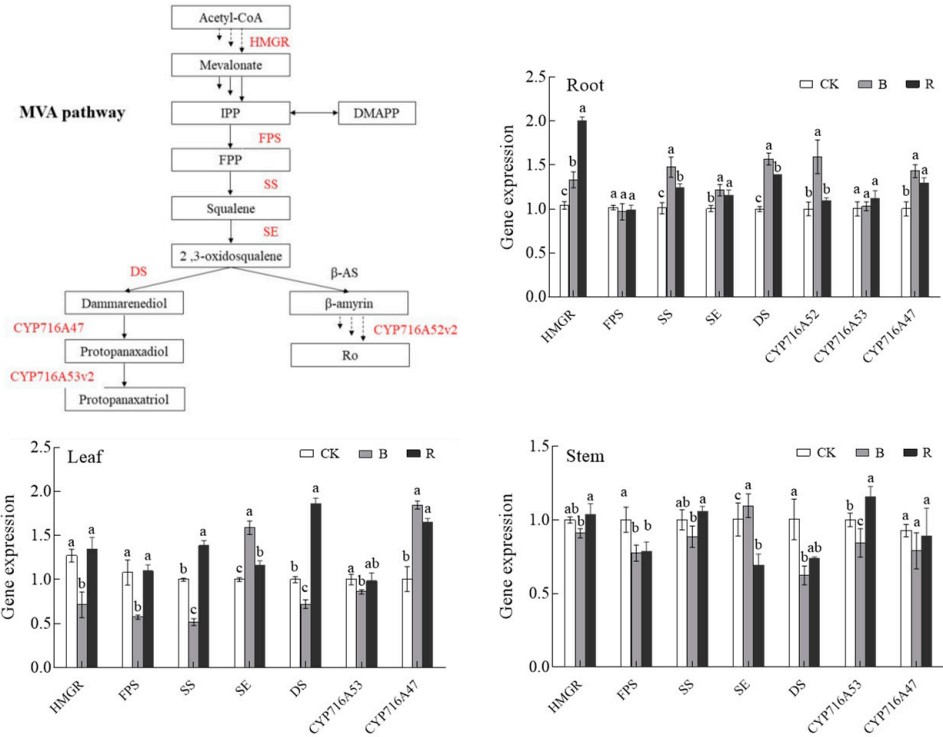

**Figure 10.** Effect of light irradiations on the gene expression quantity of *HMGR*, *FPS*, *SS*, *SE*, *DS*, *CYP716A52*, *CYP716A53*, and *CYP716A47* in *P. ginseng*; CK, white LEDs; B, blue LEDs; R, red LEDs. The data are expressed as the mean ± SD (Duncan, $p < 0.05$). The different lowercase letters indicate significant differences between treatments.

In leaves, the expression of *HMGR*, *FPS*, and *SS* was significantly lower than that of CK under blue light; however, the expression of *SE* and *CYP716A47* was significantly higher than that of CK. Under red light, the expression of *SS*, *DS*, and *CYP716A47* was significantly higher than that of CK, and the expression of other genes was less different from that of CK. The key genes involved in ginsenoside synthesis in *P. ginseng* stems were less different under different light irradiation conditions.

## 4. Discussion

LEDs have been used for vegetable and flower planting in plant factories' greenhouses. However, it is a relatively novel technology in the process of medicinal plant planting. We expected that the quality of *P. ginseng* can be improved by regulating the light environ-

ment during cultivation, subsequently promoting technological innovation in *P. ginseng* standardized planting. Therefore, we conducted a light irradiation control experiment on *P. ginseng* cultivation before *P. ginseng* harvesting. The results revealed changes in the photosynthetic physiology, secondary metabolism, and expression of key enzyme genes in the ginsenoside synthesis pathway in *P. ginseng*. The results explain the influence of different light irradiations on the synthesis and accumulation of ginsenosides, which are of great significance in improving *P. ginseng* quality at the end of *P. ginseng* cultivation.

Photomorphogenesis of plant seedlings is significantly influenced by light irradiation, red light improves plant biomass, while plants were dwarfed under blue light [27]. We regulated light irradiation at the end of *P. ginseng* growth, and the results indicated that the difference in *P. ginseng* biomass under different light irradiations was insignificant; we speculated that this was the reason why the development of organs and tissues was completed at the end of growth in *P. ginseng*, and the differences in the accumulation of photosynthetic products under the same light intensity were minute under short-term light irradiation control.

Plants perform photosynthesis by absorbing visible light from solar radiation. The photosynthetic rates of the plants changed when they were irradiated with monochromatic light of different wavelengths. Some researchers have reported that blue light is beneficial in tomato and eggplant cultivation for improving photosynthetic efficiency [41–44]. Our study discovered that white and blue light significantly increased the photosynthetic capacity of *P. ginseng*. *P. ginseng* exhibited higher Pn, Gs, *Fv/Fm*, qp, and lower Ls under white and blue light treatments. Previous studies established that blue light acts as a positive signal for stomatal opening [45]. Higher stomatal conductance may lead to the high efficiency of $CO_2$ utilization by chloroplasts, resulting in an increased photosynthetic rate of *P. ginseng* leaves under white and blue light treatments. We discovered that green light treatment resulted in a higher chlorophyll content but a weak photosynthetic capacity. Compared to red and blue light, green light is absorbed least by photosynthetic pigments; thus, green light is considered an ineffective light source [24]. The photosynthetic rate of ginseng under green light was significantly lower than that under other monochromatic light treatments. Green light can penetrate deeper into leaf tissues rather than being absorbed by the top few cell layers, such as blue light and red light [46,47]. It is established that increasing the proportion of green light in the plant spectrum is beneficial for increasing the photosynthetic rate of plants. Therefore, the reasonable use of green light in cultivation is beneficial for accumulating photosynthetic products and improving crop yield. Yu et al. revealed that red light significantly increased the photosynthetic rate of *Camptotheca acuminata* seedlings [48]; however, our study indicated that the photosynthetic rate was lower than that under white light, and NPQ significantly increased under red light. During plant leaf senescence, excess leaf light energy gradually increases NPQ levels [49]. It is speculated that *P. ginseng* enters the wilting stage early, resulting in impaired photosynthetic capacity of the leaves. The study revealed that the photosynthetic rate of *P. ginseng* leaves was lower under natural light; this was because the environmental temperature decreased in September, and *P. ginseng* leaves suffered from chilling injury, which decreased photosynthetic capacity and chlorophyll content. The effect of light irradiation on *P. ginseng* photosynthetic physiology and its regulatory mechanisms requires further research, which is relevant for *P. ginseng* light control.

Plants receive light signals through photoreceptors and transmit them downward to regulate various physiological, ecological, and secondary metabolic processes. Blue light promoted the accumulation of secondary metabolites (total flavonoids and polyphenols) in *Anoectochilus roxburghii* [50]. Blue light increased the content of total phenolic compounds and hypericins in *Hypericum perforatum*; however, red light favored biomass [51]. Our study revealed that the saponin content in *P. ginseng* roots was higher under blue and red light treatments. The total saponin content of *P. ginseng* increased by 28.81% under blue light compared to white light, and by 21.64% under red light. The biosynthesis of ginsenosides is regulated by the expression of key enzymes in *P. ginseng* [52]. To acquire

more detailed information on the regulatory mechanism of ginsenoside biosynthesis under different light irradiations, the expression levels of *HMGR*, *FPS*, *SS*, *SE*, *DS*, *CYP716A53*, *CYP716A47*, and *CYP716A52* were measured using real-time PCR. We discovered that exposure to different light conditions resulted in different transcript levels of the genes related to ginsenoside biosynthesis. The expressions of *HMGR*, *SS*, *SE*, *DS*, *CYP716A52*, and *CYP716A47* were significantly upregulated under blue light; the expressions of *HMGR*, *SS*, *SE*, *DS*, and *CYP716A47* were significantly upregulated under red light. Studies have shown that plant photoreceptors respond to different light qualities, causing a number of changes in physiological and metabolic processes. Phytochromes (PHYs) are photoreceptors of red light in plants and cryptochromes (CRYs) are photoreceptors of blue light [53]. The transcription factor HY5 is a downstream signal of photoreceptors. In *Artemisia annua*, AaHY5 activates the expression of AaWRKY9 by binding to its promoter, and AaWRKY9 contributes to jasmonate-mediation to regulate the biosynthesis of artemisinin [54]. It was hypothesized that the increase in key enzyme gene expression during ginsenoside synthesis in *P. ginseng* roots under blue and red light resulted in an increase in ginsenoside content. In addition, the expression of these key enzyme genes may be regulated by the process of light signal transduction. There may be other mechanisms behind saponin content increasing under red light. Saponins are carbon-based metabolites, and studies have shown that the excess carbon produced during photosynthesis can enter the synthesis of secondary metabolite skeletons [55]; therefore, the higher photosynthetic product content under red light may provide a substrate for saponin synthesis.

## 5. Conclusions

Our study revealed that different light irradiations changed the physiological traits and processes of secondary metabolism in *P. ginseng*. The blue light and CK treatments significantly increased the photosynthetic levels of *P. ginseng* leaves. Conversely, the photosynthetic process of ginseng leaves was restricted under green and natural light treatments. Red light exhibited a more significant effect on the accumulation of photosynthetic products in *P. ginseng*. Blue and red light increased saponin content and upregulated the expression of ginsenoside synthesis genes (*HMGR*, *SS*, *SE*, *DS*, *CYP716A47*) in *P. ginseng* roots. We speculated that blue and red light contributed to the increased ginsenoside content in roots by promoting ginsenoside synthesis (in roots) and transport (from the stem and leaf to the root), as detected in the stem and leaf ginsenoside content analyses. In summary, adopting blue or red light irradiations or covering the plants with colored films improves the quality of *P. ginseng* in greenhouses and field cultivation. The effects of different light irradiation on the morphogenesis of *P. ginseng* should be explored, which is important for the study of *P. ginseng* yield formation mechanism under the regulation of artificial light.

**Author Contributions:** L.Y. (Limin Yang) and L.Y. (Li Yang) conceived and designed the study. P.D. performed the experiments and prepared the manuscript; Z.S., L.C. and M.H. completed the experimental data processing. All authors have read and agreed to the published version of the manuscript.

**Funding:** This study was funded by the major science and technology projects of Jilin Province (20200504002YY) and the National Modern Agricultural Industrial Technology System Fund Project (grant number: CARS-21).

**Institutional Review Board Statement:** Not applicable.

**Data Availability Statement:** The original contributions presented in the study are included in the article, further inquiries can be directed to the corresponding author.

**Acknowledgments:** Thanks to Mei Han and Limin Yang, Jilin Agricultural University, for the financial support for a part of this work.

**Conflicts of Interest:** The authors declare no conflict of interest.

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
