# Peer review of "LED Light Irradiations Differentially Affect the Physiological Characteristics, Ginsenoside Content, and Expressions of Ginsenoside Biosynthetic Pathway Genes in Panax ginseng"

_agriculture, doi:10.3390/agriculture13040807_

Round 1
Reviewer 1 Report
The authors investigated the effect of light on Physiological Characteristics, Ginsenosides Content And Expressions Of Ginsenosides Biosynthetic Pathway Genes In Panax ginseng.
The article is well structured, the introduction provides sufficient background, and the materials and methods are described in sufficient detail. There are some small remarks:
Line 138 justify the choice of photosynthetic photon flux density
It would be nice to give spectrograms of light sources (NL, control treatment).
Line 187 fix the subscripts. Check throughout the text.
The symbols in figures 1-3 should be made larger.
Please indicate directions for further research.
The novelties of this paper over other relevant literature can be further highlighted.
Add DOI citations to the references.
Reviewer 2 Report
This manuscript is devoted to the study of the effect of irradiation of ginseng with various light on its physiological and biochemical parameters. Although this topic is not new for science and agriculture, there is still no unambiguous opinion on this problem. The authors conducted a large amount of experimental research. The advantage of the conducted experiments is, in my opinion, that the authors were able to provide the same illumination for all the radiation sources used.
There are questions and comments about the Manuscript:
1. How does white light (CK) differ in spectrum from natural light (NL)? It is necessary to give the spectral characteristics of all radiation sources used in the experiment. In addition, when analyzing the results, it is necessary to take into account that the sources R, G, B are monochromatic, and CK and NL have a wider spectrum.
2. Poor quality of Figure 7(a).
3. Lines 431-435 should be removed from the Discussion section.
Reviewer 3 Report
- Provide the spectrum of the white, blue, red, and green light spectrum in the supplementary material.
- Line No. 140: Explain more clearly. Why you had used two temperature25 ℃/20 ℃?
- Hyphen must be changed to minus sign throughout the manuscript. For example “-80” to “−80”
- Line No. 177 changed to 1.0 g.
- The annealing temperature for all those primers is 55 ℃.
- Provide the details of how much replication was carried out for each experiment whether you did biological replicates or technical replicates.
- Section 3.7 mention the eigenvector value of some of the metabolites
